# To Predict, Prevent, and Manage Post-Traumatic Stress Disorder (PTSD): A Review of Pathophysiology, Treatment, and Biomarkers

**DOI:** 10.3390/ijms24065238

**Published:** 2023-03-09

**Authors:** Ghazi I. Al Jowf, Ziyad T. Ahmed, Rick A. Reijnders, Laurence de Nijs, Lars M. T. Eijssen

**Affiliations:** 1Department of Psychiatry and Neuropsychology, School for Mental Health and Neuroscience (MHeNs), Faculty of Health, Medicine and Life Sciences, Maastricht University Medical Centre, 6200 MD Maastricht, The Netherlands; 2Department of Public Health, College of Applied Medical Sciences, King Faisal University, Al-Ahsa 31982, Saudi Arabia; 3European Graduate School of Neuroscience, Maastricht University, 6200 MD Maastricht, The Netherlands; 4College of Medicine, Sulaiman Al Rajhi University, Al-Bukairyah 52726, Saudi Arabia; 5Department of Bioinformatics—BiGCaT, School of Nutrition and Translational Research in Metabolism (NUTRIM), Faculty of Health, Medicine and Life Sciences, Maastricht University, 6200 MD Maastricht, The Netherlands

**Keywords:** stress, traumatic stress, PTSD, behaviour changes, pathophysiology, public health, biomarkers, prevention, treatment

## Abstract

Post-traumatic stress disorder (PTSD) can become a chronic and severely disabling condition resulting in a reduced quality of life and increased economic burden. The disorder is directly related to exposure to a traumatic event, e.g., a real or threatened injury, death, or sexual assault. Extensive research has been done on the neurobiological alterations underlying the disorder and its related phenotypes, revealing brain circuit disruption, neurotransmitter dysregulation, and hypothalamic–pituitary–adrenal (HPA) axis dysfunction. Psychotherapy remains the first-line treatment option for PTSD given its good efficacy, although pharmacotherapy can also be used as a stand-alone or in combination with psychotherapy. In order to reduce the prevalence and burden of the disorder, multilevel models of prevention have been developed to detect the disorder as early as possible and to reduce morbidity in those with established diseases. Despite the clinical grounds of diagnosis, attention is increasing to the discovery of reliable biomarkers that can predict susceptibility, aid diagnosis, or monitor treatment. Several potential biomarkers have been linked with pathophysiological changes related to PTSD, encouraging further research to identify actionable targets. This review highlights the current literature regarding the pathophysiology, disease development models, treatment modalities, and preventive models from a public health perspective, and discusses the current state of biomarker research.

## 1. Introduction

Post-traumatic stress disorder (PTSD) is a chronic mental disorder resulting in a reduced quality of life and increased economic burden. Exposure to a traumatic stressor is the trigger for PTSD development [1]. For that, a distinction between ordinary and traumatic stressors (those that have the potential to result in PTSD) is necessary. PTSD was first introduced in the Diagnostic and Statistical Manual of Mental Disorders (DSM-III), and further updates to the diagnostic criteria have been introduced in subsequent versions. Traumatic stress relates to the exposure to real or threatened injury, death, or sexual assault. Intrusion symptoms, avoidance/numbing, hyperarousal, sensitisation to stressors, and detrimental cognitive and affective changes are all symptoms of PTSD [1].

Although exposure to stressors is common in the general population, only a small proportion of susceptible individuals develop PTSD [2]; however, the underlying mechanism of susceptibility and resilience is still unclear. In the last decade, etiological models have been developed to explain the interplay between biology, environment, and mind in manifesting the disease. Examples of those models include the diathesis–stress and the biopsychosocial models [3]. In parallel, extensive research aiming to identify the pathophysiological mechanism of PTSD has found the association between genetic variants and an increased risk of PTSD, hypothalamic–pituitary–adrenal (HPA) axis dysfunction, neurotransmitter dysregulation, and alterations in brain circuits [4]. Research has advanced over the last years in aiming to connect the alterations at the genetic, molecular, chemical, cellular, and circuitry levels into a biological systemic view, while also aiming to discover diagnostic and prognostic biomarkers.

Psychotherapies and pharmacotherapies are two effective PTSD treatments. However, significant subsets of individuals who do seek treatment have symptoms that are difficult to treat. Changing to another treatment modality or combining treatment modalities (combining psychotherapy with pharmacotherapy) is frequently required. Trauma-focused psychotherapy is the first line of treatment for most individuals with PTSD, as opposed to other therapies or pharmacological medication. Cognitive behavioural therapy (CBT), prolonged exposure therapy (PET), and eye movement desensitization and reprocessing (EMDR) therapy are trauma-focused psychotherapies that have been shown to be useful in the treatment of PTSD [5,6].

This review provides an introductory and overall overview of the current concept of findings on the etiology and disease models of PTSD, pathophysiology, treatment, prevention, and lastly biomarkers, including diagnostic and prognostic biomarkers. A literature search was performed using keywords to find papers in PubMed, Cochrane Library, Scopus, and Embase. The literature was then summarized with the aim to provide a comprehensive overview of these topics, and representative examples of research findings were selected. The preferred studies were meta-analyses, systematic reviews, and randomised controlled trials (RCTs), as well as the most recent studies relevant to the presented perspective. Abstracts were then screened for their relevance. Once selected, the limitations of the studies were assessed for their impact on informed clinical decisions. Original studies were preferred for models and concepts. The literature was then summarised with the aim to provide an up-to-date, comprehensive overview of the available data (a graphical summary of the main findings is given in Figure 1).

## 2. Epidemiology and Models of PTSD Development

### 2.1. Epidemiology of PTSD

The public health perspective of traumatic stress takes a population-based approach and formulates policies based on it. Epidemiological studies concern the distribution and determinants of traumatic stress and stress-related mental disorders in specified populations. They have shown that traumatic stress often occurs among the war-surviving population, refugees, especially females, public health workers, and indigenous populations [7].

The war-surviving population has a high prevalence of PTSD and depression. This statement is supported by a systematic literature search and meta-analysis of interview-based epidemiological surveys including samples from 43 war-ridden countries with a recent war history (1989–2019) [8]. Specifically, these war survivors diagnosed with PTSD or depression are primarily living in low/middle-income countries [8]. Therefore, income or social status may have an impact on the response to traumatic stress. A meta-analysis and the Millennium Cohort study support the association between low socioeconomic status and PTSD [9,10]. Sex/gender has also been linked to risk, e.g., in a refugee population, females who experienced sexual trauma had a higher prevalence of PTSD than males [7].

Differences between groups of people have been reported. For example, indigenous populations in various countries show a higher prevalence of traumatic stress-related mental health problems than others. The standardized prevalence of 12-month PTSD in the Australian indigenous population was three times the Australian rates [11]. The independent predictors (determinants) of PTSD among Australian indigenous populations are female gender, rural residence, trauma under age 10, and sexual and/or physical violence [11]. Such findings might be attributed to the fact that the Australian indigenous population is disadvantaged in different aspects [12].

Public health workers have been experiencing huge traumatic stress during the COVID-19 pandemic [13]. Among 26,174 surveyed public health workers in the US, 53.0% reported symptoms of at least one mental health condition in the previous two weeks, especially those unable to take time off or those experiencing overwork [13]. This result indicates that overworking is associated with the negative impact of traumatic stress.

The studies above demonstrate that specific populations with certain characteristics are more likely to suffer from exposure to traumatic events. These characteristics can be environmental factors like traumatic events including war/violence and workload, social factors like social status, or biological factors like female sex. A dual influence of social and biological factors on females is suggested. Females are more likely to experience physical and sexual violence. Additionally, clinical evidence points to the possibility that cyclical oestrogen discharges throughout the reproductive cycle may contribute to women’s greater susceptibility to and severity of PTSD symptoms following psychological stress [14,15].

### 2.2. Models of Disease Development

#### 2.2.1. Diathesis–Stress Model for PTSD

The initiation and maintenance of the disorder are heterogeneous, differing between individuals with the same level of trauma and presenting with varying degrees of disease severity. An explanation for this obvious variation is the diathesis–stress model, tested for PTSD in different studies. According to the model, the pre-trauma state of the individual (risk factors) constitutes a condition of susceptibility (diathesis) that can produce the disorder after a traumatic experience (stress).

In PTSD, the trauma represents the stressor that activates certain processes in an individual with pre-traumatic vulnerability, thereby leading to the expression of psychopathology and social dysfunction. Vulnerability factors may involve aspects like genetic predisposition, psychiatric history, a history of child abuse, a stressful and unhealthy lifestyle, and others. The individual’s diathesis represents a hypothetical threshold, and the impact of a stressor on the individual depends on the diathesis; the less favourable the diathesis, the less severe the stressor needs to be to initiate the disorder.

Not only pre-trauma (e.g., a risk of developing PTSD, genetic, and biological factors), but also peri-trauma (e.g., emotional distress), and post-trauma factors should be considered. McKeever and Huff state that the peri-traumatic perception of trauma and post-traumatic conditions can affect the severity and symptoms. Furthermore, different types of vulnerabilities (e.g., biological and psychological) may interact with one another [3].

Here, we provide a few examples of research findings in line with the diathesis–stress model. As the diathesis–stress model contends that the interplay of hereditary, biological predisposition, and environmental stress results in the development of mental disorders, researchers have hypothesized that during military service the onset of major psychopathology may be precipitated by psychosocial stress, leading to an increase in psychiatric hospitalisations during the first months of the military service period for those with greater sensitivity or a lower stress tolerance [16]. In a sample of 118 hospitalised subjects starting their military service assessed for the expression of psychopathology, 59.3% of the subjects were diagnosed with an anxiety disorder, especially PTSD, out of the total sample due to traumatic stress exposure, implicating the nature of warfare stress in the increased risk of anxiety and stress disorders. Intriguingly, the risk of disorder onset within the first two months and hospitalisation was also higher for psychotic spectrum disorders. As the sample was exposed to similar stress levels, these findings suggest that individuals with psychotic spectrum disorders have increased stress sensitivity [16].

Another study for testing the model in the emergency department (ED) was carried out by Edmondson in 2014. A sample of 189 acute coronary syndrome patients was observed for the effect of ED crowding, depression status, and their interaction on the subsequent development of PTSD. ED crowding significantly affected the 1-month development of PTSD symptoms, as patients treated during ED crowding times scored significantly higher than those treated during times with medium or little ED crowding. Similarly, depression status and the interaction between ED crowding and depression status significantly affected the subsequent development of PTSD [17].

Another example of findings supporting the diathesis–stress model of PTSD comes from Elwood and colleagues, who investigated the connection between cognitive abilities/vulnerabilities and exposure to sexual assault. Negative attributional style (NAS) and anxiety sensitivity (AS) were used as cognitive vulnerabilities and sexual assault as the negative life event. NAS is the individual’s inference that current negative events will have negative effects, and that a negative event reflects one’s worthlessness, while AS is the extent of fear of the harmful consequences that can result from anxiety and anxiety symptoms [18,19]. In line with the diathesis–stress model, the authors hypothesized that people who had both high levels of cognitive vulnerability and high levels of negative life experiences would have the highest levels of symptoms. The relation between these cognitive vulnerabilities and negative life events was examined for PTSD symptom clusters [20]. As expected, negative life events significantly predicted changes in avoidance, numbness, and dysphoria symptoms when both NAS and AS were present. The biggest symptom increase was recorded by participants who exhibited high levels of cognitive vulnerability and more traumatic life events. Correspondingly, a low frequency of bad life events and high levels of cognitive insensitivity, however, were linked to reductions in symptoms [20]. Findings from this study support the role of cognitive vulnerabilities as predictors of the development of PTSD symptoms after exposure to traumatic stressors.

#### 2.2.2. Biopsychosocial Model

Formulated by Engel in 1977, the biopsychosocial model emerged from the view of the insufficiency of the biomedical model alone in explaining illnesses [21]. Engel explained a need for a new medical model to extend the biomedicine model to account for all the factors influencing the patient’s condition. The biopsychosocial model poses that biological (e.g., genetics, chemical changes, and organ damage), psychological (e.g., stress, mental illness, behaviour, and personality), and social factors (e.g., peers, socioeconomic status, beliefs, and culture) interact with each other in the expression of health and illness. Engel’s justification for his criticism was made in several points:Biological disturbance alone is insufficient to cause the disease, as disease appearance results from multi-factor interaction.Vulnerability is better accounted for by psychological and social factors than by biological changes.The effectiveness of biological treatments is influenced by the psychological status of a “placebo effect.”Health outcomes are affected by the doctor–patient relationship to a great extent.

The biopsychosocial model would consider these factors altogether, not only in terms of the expression of illness but also at the level of the social functioning of the individual. It also considers the cultural perception of illness, circumstances in which the patient does not acknowledge their illness, and other circumstances in which the patient admits their illness, as marked by their entry into the healthcare system [22].

#### 2.2.3. Animal Models of PTSD

In 1993, Yehuda and Antelman proposed a list to systematically evaluate stress models of stress in animals for their relevance to PTSD [23]. To determine how applicable a model is to PTSD, at least five distinct factors might be applied, according to this list:(1)Even very brief stressors should cause biological or behavioural symptoms of PTSD;(2)The stressor should be able to produce symptoms in a dose-dependent manner;(3)Produced biological alterations should persist or become more pronounced over time;(4)Alterations should have the potential to express biobehavioural changes in both directions;(5)Interindividual variability in response is present as a function of experience and/or genetics.

A sixth criterion has been proposed by Whitaker et al., which is the model’s capacity to generate co-morbid states, such as an increase in alcohol consumption, an increase in compulsive drinking, and hyperalgesia [24]. Although these models do not fully replicate the human condition, they simulate the symptoms and neurobiology of PTSD, allowing the evaluation of behavioural changes, neurobiological and epigenetic alterations, and the development of biomarkers and treatment. According to the type of stressor, the models can be categorised as physical, social, or psychological. Table 1, provides a summary for the commonly used animal models in PTSD, which are mostly conducted in rodents.

## 3. Pathophysiology

While much about the pathophysiology of PTSD is unknown, research into the pathophysiological aspects of PTSD is in rapid development. Preclinical investigations of animal models of stress and evaluations of biological variables in populations with the condition have all contributed to the identification of biological factors and mechanisms involved in PTSD, which can be described on the levels of brain circuits, neurochemical factors, and HPA axis, as discussed in this section.

### 3.1. Brain Circuits

The core features of PTSD are fear and worry in conjunction with other features and symptoms, including arousal, avoidance, sleep disturbance, and intrusion symptoms (e.g., flashbacks and nightmares) [35]. Neural circuitries and biological processes underlying these features involve brain structures such as (i) the amygdala, anterior cingulate cortex, and the insula in dysfunctional threat detection; (ii) frontoparietal regions (the dorsolateral prefrontal cortex, ventrolateral prefrontal cortex, and medial prefrontal cortex) in emotional regulation; (iii) the medial prefrontal cortex and the hippocampus in contextual processing [35].

A range of structural magnetic resonance imaging (MRI) studies has reported structural abnormalities in the hippocampus and anterior cingulate cortex (ACC) in patients with PTSD [36]. Additionally, functional magnetic resonance imaging (fMRI) studies have reported increased activity of the amygdala, which processes fear and emotion, and decreased prefrontal cortex activity when completing tasks that use either trauma-related or unrelated stimuli (script-driven recollections of trauma-related and unrelated stressful events) [37,38].

While hippocampal alterations have been observed in patients with PTSD, including lower volumes and lower levels of activation [39], impaired connectivity in the frontoparietal areas, both inside and between executive function networks, has also been observed in patients with PTSD [40]. The aforementioned findings suggest that a disrupted connection within these circuits may reflect a vulnerability factor for PTSD. In contrast, people who were exposed to traumatic events but who do not develop PTSD were reported to exhibit higher prefrontal cortex activity during extinction recall [41,42], and stronger connections between the ACC and the hippocampus, compared to patients with PTSD [43].

The novel findings by Borgomaneri et al. support the idea that the dorsolateral prefrontal cortex (dlPFC) plays a crucial role in the neural network that mediates the reconsolidation of fear memories in humans by showing that non-invasive repetitive transcranial magnetic stimulation (rTMS) of the prefrontal cortex after memory reactivation interferes with the expression of fear towards a previously conditioned threatening stimulus. These results enhance our understanding of the processes behind fear memory reconsolidation, and also have potential therapeutic applications in treating fear memories [44]. Identifying the brain regions involved in the reconsolidation of emotional memories and their particular interactions within the overall fear-processing network remains a challenge for non-invasive brain stimulation (NIBS) and reconsolidation-based interventions, which are increasingly applied to conditions like PTSD [45].

### 3.2. Neurochemical Factors

#### 3.2.1. Dysregulation of the Noradrenergic System

Catecholamine noradrenaline is a critical transmitter in the autonomic nervous system, and has been linked with the development of the autonomic symptoms associated with PTSD. Noradrenaline is found in the central nervous system’s (CNS) cell nuclei and certain noradrenergic pathways that are implicated in the pathophysiology of the illness. One area with a high concentration of noradrenaline is the locus coeruleus (LC), which is located in the rostral pons and serves as the hub of the neurochemical activity associated with PTSD [46]. Clinical and preclinical evidence suggests that the dysregulation of noradrenergic signalling is involved in the pathophysiology of PTSD. The increased noradrenergic tone in PTSD arises from increased central and peripheral sympathetic activity leading to increased resting heart rates and systolic blood pressure [47]. In addition, noradrenaline levels are higher in the urine of individuals with PTSD than in healthy individuals, but recent studies have failed to establish such findings in the cerebrospinal fluid (CSF) [48,49].

Many of the symptoms of PTSD emerge from an increased CNS noradrenergic tone [49,50]. Increased noradrenaline activity has been linked with dysfunction of the medial prefrontal cortex (through impairing prefrontal signalling via α1- and β-AR in the prelimbic (PL) and infralimbic (IL) subdivisions of the medial prefrontal cortex) and disturbed fear extinction, which may underlie the increases in behavioural measures of anxiety and PTSD symptom severity [51,52]. Excessive noradrenaline release was found to be increased in the hyperactive amygdala and LC, resulting in intrusion symptoms and autonomic hyperactivity [53].

Altered noradrenergic function is also associated with night-time and sleep symptoms in PTSD. Increased sympathetic activity during sleep, e.g., an increased heart rate, is found in individuals with PTSD [54]. Additionally, during rapid eye movement (REM) sleep, people with high levels of PTSD-like symptoms showed an increase in the ratio of low-frequency to high-frequency heart rate variability, which is associated with an elevated sympathetic tone [55].

#### 3.2.2. Dysregulation of Serotonin Signalling

Serotonin (5-HT) is a monoamine neurotransmitter with multiple biological functions related to mood, cognition, memory, and behavioural regulation [56]. The 5-HT signalling in the amygdala has been linked to fear regulation and threat responsiveness. Several 5-HT receptors, including 5-HT_1A_, 5-HT_1B_, 5-HT_2A_, and 5-HT_2C_ have been linked to PTSD and anxiety [57,58].

According to reports from pharmacological studies, blocking the serotonin 5-HT2C receptor in rodents increases locomotion and reduces anxiety [59,60], and in addition, the 5- HT_1A_ receptor agonist induces anxiogenic responses to the elevated plus maze (EPM) test in mice [61]. A recent study found that both 5-HT_1A_ and 5-HT_2A_ in the hippocampus mediate anxiety-like behaviour in a mouse model of PTSD via the ERK pathway [62]. Clinical evidence showed higher 5-HT_1A_ binding potential in people with PTSD, particularly in those with comorbid MDD [63].

#### 3.2.3. Dopamine

Another prevalent neurotransmitter in the brain is dopamine, primarily synthesised in midbrain areas [64]. Dopamine is a neurotransmitter that is involved in the regulation of motor activity, limbic functions, attention, and cognition, particularly executive function and reward processing [65,66]. It makes a significant contribution to the anticipatory processes required for planning voluntary action after intention as well as behavioural adaptability [67]. A range of studies have investigated links between the dopaminergic system and PTSD. For example, several studies have attempted to link PTSD with genetic variants in certain dopamine receptor genes (e.g., DRD2) [68,69,70]. Other studies have focused on dopamine-beta-hydroxylase (DBH), which catalyses the conversion of dopamine to noradrenaline, and have reported that high-plasma DBH levels may be linked to the development of intrusion symptoms [71,72], yet other studies have focussed on the dopamine transporter SLC6A3 (solute carrier family 6, member 3), a member of the sodium- and chloride-dependent neurotransmitter transporter family which mediates the transport of dopamine from the synaptic cleft. The 9R allele of the SLC6A3 locus has been identified as a risk allele for PTSD [73]. Additionally, the epigenetic state of the promoter region of SLC6A3 has been identified as a potential risk factor for/indicator of PTSD [74].

#### 3.2.4. Gamma-Aminobutyric Acid (GABA)

The inhibitory neurotransmitter GABA is widely distributed throughout the entire brain. A complex pattern of results has been found in studies comparing GABA levels in numerous brain areas between those with and without PTSD. A proton magnetic resonance spectroscopy (MRS) study reported lower GABA-levels in the temporal cortex, parieto-occipital cortex, and insula and higher GABA levels in the dorsolateral prefrontal cortex in people with PTSD compared to trauma-exposed healthy controls [75]. In times of extreme stress, low plasma levels of GABA are related to PTSD and may lead to the overload of hyperadrenergic response regulation [76].

There are three primary classes of GABA receptors: GABA-A, GABA-B, and GABA-C. Human research has revealed that Vietnam War combat veterans with PTSD had lower GABA-A benzodiazepine binding ability. According to these findings, changes in the GABA receptor’s expression or binding ability may have an impact on mental diseases linked to stress, including PTSD [77,78].

#### 3.2.5. Neuropeptide Y (NPY)

NPY is a neuropeptide that is expressed throughout the brain, including the forebrain, limbic system, and brainstem. It is involved in several physiological processes including the regulation of emotional and stress-related behaviours [79].

Early research developed a concept that NPY counteracts the actions of the corticotropin-releasing factor (CRF), terminating the stress response and countering the HPA axis [79,80,81]. Studies have demonstrated that plasma NPY levels rise in response to stress, and that higher NPY levels are associated with better behavioural performance under stress [82,83].

Plasma NPY levels were assessed in soldiers who took part in a survival course meant to mimic the conditions that prisoners of war would encounter. Within a few hours of being exposed to military interrogations during the survival course, their serum NPY levels increased. Furthermore, compared to the non-Special Forces or regular soldiers, the majority of Special Forces members who had received resilience training had much higher NPY levels [84].

Several preclinical and clinical research reports point to an association between PTSD and decreased NPY in the CNS [79,85]. Moreover, NPY levels appear to increase after PTSD remission, suggesting that NPY may act as a biomarker of PTSD or at the very least as a resilience element [86].

Together, these investigations show that NPY levels in PTSD patients closely mirror the disease course and that NPY can operate as a stress buffer in response to stressful experiences by lowering noradrenergic hyperactivity [87]. The most prevalent SNP for NPY investigated is the rs16147 (399T > C) polymorphism, which is linked to low levels of NPY and has been linked to hyperarousal, changes in the HPA axis response to stress, and the activation of the hippocampus and amygdala [79]. Another NPY SNP is 1002T > G, associated with low NPY content in the CSF and amygdala, which is linked to higher levels of anxiety, arousal, addictive behaviours, and decreased stress resilience [79].

#### 3.2.6. Brain-Derived Neurotropic Factor (BDNF)

BDNF, the richest neurotrophin in the brain, was first characterised for its involvement in the formation of the CNS. It has the ability to participate in neural activities such as survival, differentiation, development, and neuronal plasticity. It preserves synaptic plasticity, which is necessary for extinction learning and fear memory storage [88]. Variations in BDNF expression or in genetic background have been linked with a risk of various psychiatric disorders such as anxiety, depression, and PTSD [89].

In US military personnel deployed throughout the conflicts in both Iraq and Afghanistan, PTSD patients had higher BDNF protein levels in their peripheral blood plasma than non-PTSD controls. In the inescapable tail shock rat model of PTSD, increased BDNF levels were found in both blood plasma and hippocampus tissue. Furthermore, the polymorphism Val66Met in BDNF has been linked with an increased risk of PTSD, exaggerated startle response, and alterations in fear extinction [90,91]. The same polymorphism was also found to affect hippocampal volume and memory [92]. Together, these findings emphasise that BDNF and related molecules may be interesting candidates for biomarker studies and for more fundamental studies aiming to identify actionable biological targets related to the onset or course of PTSD [89,93].

#### 3.2.7. Cannabinoid and Opioid Receptors

Endogenous cannabinoids, including anandamide (AEA) and 2-arachidonolyflycerol (2-AG), work via cannabinoid (CB) receptors (CB1R, CB2R), which are implicated in the pathogenesis of PTSD [94]. Preclinical data showed that AEA levels are decreased in the brain in chronic stress models [95]. This is in agreement with the human data showing that endocannabinoid plasma levels are reduced in PTSD patients [57,96]. In addition, defective endocannabinoid signalling is correlated with glucocorticoid dysregulation associated with PTSD [96].The CB1 receptors are the most abundant G-protein-coupled receptors in the CNS and are highly expressed in a fear circuit of the cortical and subcortical brain regions associated with PTSD [97]. Interestingly, the disruption of the CB1 receptor gene (knockout models) was found to increase anxiety, whereas a pharmacological blockade of the receptor had anxiolytic effects [98,99]. Animal stress studies also showed that CB1 receptor expression was increased in female but not male animals [100,101].

#### 3.2.8. Oxytocin

Oxytocin is a neuropeptide produced in hypothalamic periventricular and supraoptic nuclei. It emerges from the posterior pituitary and enters the bloodstream. Oxytocin is transported to different parts of the brain through neuronal projections from the hypothalamus. The amygdala, brainstem, olfactory nucleus, and anterior cingulate cortex are among the human brain areas that express the oxytocin receptor and are therefore likely to be impacted by oxytocin [102]. As it acts on brain areas involved in PTSD, and as oxytocin appears to have anti-stress and anxiolytic effects, oxytocin is thought to be involved in the constellation of dysregulations found in PTSD [103,104].

### 3.3. Dysfunctional HPA Axis

The hypothalamus, pituitary, and adrenal glands make up the HPA axis, a hierarchical system that controls how the body responds to stress from the environment while maintaining homeostasis [105,106]. As the axis is one of the main stress response systems that controls the release of cortisol and stress hormones, it has received a lot of attention. Exposure to stress causes increased corticotropin-releasing hormone (CRH) production from the hypothalamus [107,108]. The HPA axis is first activated by the production of CRH, which travels through the infundibular stalk’s hypophyseal portal arteries to the anterior pituitary, where it binds to CRH type 1 receptors (CRF1) to trigger the release of adrenocorticotropin (ACTH) into the bloodstream. Cortisol, the main HPA axis effector chemical, is released when ACTH binds to melanocortin 2 receptors in the zona fasciculata of the adrenal cortex. In order to promote the stress response, cortisol has a number of physiological impacts throughout the body, including blocking insulin signalling and increasing glucose availability, controlling immune system operations, and altering electrolyte balance [109,110,111].

Upon CRH administration, rodents exhibit PTSD-associated behaviours [112]. In addition, CRF-1 knockout mice showed impaired responses to stress and reduced anxiety [113,114]. At the same time, CRF-2 knockout mice showed hypersensitivity to stress and augmented anxiety [113,115]. PTSD patients have high CSF levels of CRH and a dysfunctional HPA axis [107,116,117]. Studies indicate CRH hyperactivity with subsequent glucocorticoid receptor (GR) hypersensitivity, resulting in higher negative feedback inhibition of cortisol and CRH release [118,119]. In a meta-analysis, Morris et al. reported significantly lower basal cortisol levels in PTSD and trauma-exposed controls without PTSD compared to non-traumatised individuals. Additionally, individuals who had experienced childhood trauma had significantly lower morning cortisol levels compared to those exposed to adulthood trauma [120]. Figure 2 provides a general overview of the information about the HPA axis discussed in this section.

### 3.4. Conclusive Remarks

Despite the extensive discoveries, the current understanding of the neurochemical factors in PTSD is still limited and requires more research. There is a number of understudied yet significant subjects in the discipline, such as variables that affect susceptibility and resiliency. For instance, one such subject is whether or not the exogenous administration of oxytocin and NPY, two neurobiological components that protect against stress, can foster resilience. Additionally, determining relationships between heritable variables (genetics and epigenetics) and trauma exposure is crucial to understanding PTSD risk, and predicting treatment response. It is important to thoroughly evaluate how trauma affects gene expression, neural plasticity (across the CNS), circuit remodelling, and neurotrophic factors. Future studies should focus on the characterisation of proteomic and transcriptomic abnormalities in PTSD, with the integration of GWAS and EWAS studies, in order to map out novel networks, and allow the development of reliable biomarkers.

Likewise, current controversies and conflictions in the studies assessing HPA axis function and diurnal cortisol levels can be a result of the methodological heterogeneity and limitations of the studies (e.g., different methods of cortisol measurement, different timings of cortisol measurement, and different methods in establishing PTSD in addition to other statistical limitations). Further studies with greater homogeneity are required to draw definite conclusions.

## 4. Prevention Model of PTSD

PTSD has a predictable development pattern and follows a specific triggering event, unlike other mental diseases. Early PTSD symptoms appear days after exposure to stress. Emergency care providers and helpers are made aware of a lot of traumatised people. These circumstances present exceptional chances for identifying those who are in danger and offering preventive measures. Despite these benefits, the effective prevention of PTSD remains challenging, and the disorder’s incidence in both military members and civilians over the past decades has been relatively steady [121]. With enough understanding of the condition, preventive and interventional measures can be used to enhance quality of life and reduce the disease’s financial and medical burdens. This is supported by the development of prevention models, but also by the modern digital support of their implementation by e-health approaches (the latter will be explained in the treatment section).

### The Social-Ecological Model for PTSD Prevention

A social-ecological model as a framework for prevention is proposed by the US Centers for Disease Control and Prevention, which emphasises risk factors at multiple levels, including the individual level, relationship level, community level, and societal level [122] (Figure 3). Risk factors on the individual level are about personal characteristics, which are the model’s core. Furthermore, the risk factors on the relationship level are about the quality of relationships for the individual in the family, friends, or other interpersonal interactions. In addition, the risk factors on the community level could be the feeling of safety and economical status. Lastly, the risk factors on the societal level could be social norms, cultural background, and tolerance [123].

Therefore, preventive measures on the individual and relationship level should be individualised and personal. For instance, emergency hotlines and psychological counselling services should be available in the way that is easiest to reach. Accordingly, preventive measures on the community and societal level are public, legitimacy-related, and educational. For instance, the community or public health sector should increase the awareness of the impact of traumatic stress, and it could organise lectures, campaign movements, or give out brochures about traumatic stress, first-aid help information, and preventive measures (Figure 3). Community-based interventions to improve mental health for people in low- and middle-income countries usually use lay community members as intervention deliverers, and apply transdiagnostic approaches and customized outcome assessment tools [124]. Furthermore, the public sector should formulate laws, legislation, and policies to prevent discrimination and racism to protect specific populations [125].

Additionally, preventive measures should also be taken during different stages of traumatic exposure [123]. Before exposure, the primary prevention is of the actual occurrence of disease or illness. After stress exposure, the secondary form of prevention is to intervene early in the disease process for a cure, and for the reversal of illness or for optimal outcomes. When a disorder occurs, tertiary prevention steps are taken to prevent the disability that often accompanies an illness or disease [123] (Figure 3). For optimal outcomes, primary, secondary, and tertiary prevention measures should be evidence-based as they are part of disease management [126].

## 5. Treatment Modalities for PTSD and E-Mental Health

PTSD is often a chronic and disabling disorder. Many patients fail to seek medical care, and others have symptoms resistant to treatment. Early treatment as soon as the diagnosis is made is recommended to prevent chronicity and disability [127]. The main goal of treatment is to improve quality of life, maintain patient and others’ safety, reduce distressing symptoms, and reduce hyperarousal and avoidant behaviours. The first-line intervention for PTSD patients is psychotherapy, either trauma-based psychotherapy (CBT, exposure therapy (ET), or EMDR) or non-trauma-focused psychotherapy (present-centred therapy, interpersonal therapy, or mindfulness therapy). In the case of psychotherapy failure, pharmacotherapeutic treatment options are the next choice. Additionally, if the patient has a disability that impairs the success of trauma-focused psychotherapy, pharmacotherapy is considered the appropriate choice until psychotherapy can be initiated [5,6]. In addition to current therapies, e-health is gaining attention as it has interesting potential for providing training, assessment, prevention, and the treatment of negative effects after trauma exposure on a global scale [128].

### 5.1. Trauma-Based Psychotherapy

As mentioned earlier, trauma-based psychotherapy includes CBT, ET, and EMDR. CBT has a cognitive and behavioural component. The cognitive component is mainly focused on the cognitive reconstruction of the effects of a traumatic event on one’s life, by addressing all maladaptive beliefs and thoughts about safety, power, trust, and control, while the behavioural component is about learning how to deal with and challenge these thoughts through thinking or real experience, in order to achieve symptom reduction [129].

ET can be imaginal exposure, in vivo exposure, or virtual reality exposure. All of these focus on putting the patient in confrontation with their traumatic event and memory for these to become less distressing. In PET, multiple sessions of education on reactions to trauma, processing traumatic material, and breathing training are undertaken. It was shown to be effective in patients with comorbid conditions such as psychosis, personality disorders, and substance use [130,131]. In written ET, the patient writes about their traumatic events in response to certain stimulations and discusses them with the therapist to pay attention to the thoughts and events that evoke the patient’s symptoms, with exposure being imaginal [132].

EMDR is a combination between cognitive behavioural therapy and ET in addition to saccadic eye movements during the therapy. The patient remembers the traumatic event, and while focusing on the cognition aspects simultaneously, the therapist moves their fingers in front of the patient and asks the patient to follow them repeatedly until the anxiety subsides [133].

### 5.2. Non-Trauma Focused Psychotherapy

This approach includes present-centred therapy, interpersonal therapy, and mindfulness-based stress reduction. Present-centred therapy focuses on the current life stressors and how to cope with them [134]. Interpersonal therapy focuses on a specific symptom and impairment in the context of interpersonal relationships [135]. Mindfulness-based stress reduction mainly teaches the patient how to be fully focused on the current moment, not think about the traumatic event, and attend to the present in a non-judgmental manner [136].

### 5.3. Pharmacological Therapy

The pharmacotherapy treatment is preferred in the case of psychotherapy treatment resistance, different patient preferences, or a patient’s inability to participate in the former, and mainly comprises selective serotonin reuptake inhibitors (SSRI) and serotonin-noradrenaline reuptake inhibitors (SNRI). Occasionally, second-generation antipsychotics (SGAs) such as risperidone or olanzapine can be used. If effective, pharmacotherapy should be continued for at least six to twelve months to prevent relapse [137].

Due to their efficacy at reducing symptoms of PTSD, SSRIs and SNRIs are the first-line agents in the pharmacotherapy of PTSD. Treatment with SSRIs resulted in a higher reduction in the Clinician-Administered PTSD Scale (CAPS) score than treatment with a placebo in a meta-analysis of 12 studies including 1909 PTSD patients, with paroxetine and sertraline being most effective among SSRIs [138]. The approach to the usage of SSRIs is to “start low and go slow” until the response is achieved, in order to avoid unwanted side effects. However, failure cannot be determined until the maximum therapeutic dose is given and a period of 6–8 weeks is completed, with at least two different agents being used before documenting failure [139,140,141]. Although few studies have compared SSRIs to SNRIs, randomised trials have indicated that venlafaxine was superior to a placebo in decreasing PTSD symptoms [142]. Second-generation antipsychotics can be used as a monotherapy or as augmentation therapy in the case of concomitant psychosis or in the case of failed SSRI/SNRI [127,143,144]. In a trial involving 247 United States military veterans from the US who did not respond to two or more trials of SSRI and SNRI, participants received either 4 mg of risperidone or a placebo. However, no significant difference was observed between the two groups between the CAPS scores [145]. In another study, eighty United States military veterans with persistent PTSD were given quetiapine monotherapy as opposed to a placebo in a randomised clinical study. After 12 weeks, the individuals who received quetiapine had higher mean reductions in their CAPS total score than those who received a placebo [146]. RCTs and systematic reviews conducted on other SGAs (i.e., olanzapine, aripiprazole) show that these agents are reasonable either as monotherapies or augmentation therapies [147,148].

The alpha-1 blocker prazosin is mainly used for symptom relief, especially during sleeping. It is the preferred agent in patients experiencing nightmares or sleeping disorders. It can be used as an adjunct to SSRI/SSNRI [149]. There are some preliminary clinical studies on riluzole (a glutamatergic modulator), 3,4-methylenedioxymethamphetamine (MDMA), and ketamine. The results of these studies are promising, but these agents are not yet approved [150,151,152]. As there is no clear benefit, benzodiazepines are not recommended, as they have been shown to decrease the effectiveness of psychotherapy by diminishing the extinction effect, and they should be avoided, especially in patients with substance use disorders [153,154]. The effectiveness of anticonvulsant drugs with mood-stabilising effects to lessen PTSD symptoms has not been adequately explored in clinical studies. Few sufficiently powered, randomised trials have been released, and the results have largely been unfavourable [155,156,157,158]. On the other hand, intranasal oxytocin administration has been found to be a promising approach to preventing and reducing PTSD symptoms. Frijiling et al. found that repeated intranasal oxytocin administration in the early post-trauma period reduced the development of PTSD symptoms [159]. Additionally, intranasal oxytocin was found to reduce symptom severity in females with PTSD upon a trauma-script challenge [160]. Oxytocin has also been suggested to enhance the outcomes of psychotherapy, although adequately powered RCTs are still needed to assess this use of oxytocin [161].

A novel method of treating PTSD involves the disruption of memory reconsolidation [162]. Pharmacologically, this could be done through trauma memory reactivation with the administration of an amnesic drug, resulting in the disruption of memory consolidation. One drug showing promise in such an approach is propranolol, as it has been shown to disrupt fear memory reconsolidation in the amygdala in rodents [163]. Despite the limitations, such as short-term follow-up and the conflicting results of the recent studies, propranolol shows promise as an early preventative measure for PTSD.

### 5.4. E-Mental Health and Virtual Reality (VR)

To integrate our understanding of traumatic stress as a public health problem, interdisciplinary and modern approaches can facilitate the mission, including health service research [164], internet-based digital approaches like e-health [165], and artificial intelligence applications like virtual reality (VR) [166].

In the area of traumatic stress, e-mental health, defined by Riper et al. as “the use of information and communication technology to support and improve mental health conditions and mental health care”, has enormous potential to provide instruction, assessment, prevention, and treatment for negative effects following trauma exposure globally [167]. It is possible and effective to give intensive, trauma-focused treatment for severe or complicated PTSD via home-based telehealth. This can be a substitute for trauma-focused treatment that is delivered in person [168]. E-health being offered in times of pandemics (such as with COVID-19) has shown to be an efficient way to support prevention and possibly intervention at times of shortage [169]. This suggests that also in global PTSD care, this could be effective. However, several challenges arise in convincing patients to undergo such a new method of care. Bakker et al. propose three ways in which the application of e-health can be accelerated. First, optimising adherence and the engagement of users (including patients, clinicians, and relatives) can be achieved by designing approaches that meet the requirements of the users and implementing a holistic approach instead of focusing on a single disorder, in addition to featuring designs that engage the patients, such as real-time engagement, rewarding systems, and the involvement of VR and augmented reality (AR) programs. Second, increasing the field’s research with clinical and high-quality studies to help test evidence-based medicine for effective interventions. Lastly, the wide implementation of such interventions could be of great benefit, especially when local expert clinicians and clinics are unavailable. However, such a wide implementation must overcome several dilemmas, including physician and patient avoidance of internet interventions and patients’ preference of therapist-guided interventions [165].

VR for ET or psychophysiological assessment and resilience training could reduce negative impacts or enhance well-being in response to traumatic stress in public health [166]. Studies show the promising and wide usage of VR in trauma management from combat scenarios to the COVID-19 pandemic [166].

### 5.5. Conclusive Remarks

Although recent developments utilising methodologically sound designs have increased confidence in the effectiveness of PTSD therapy, a sizable proportion of patients fail to respond to treatment, stop receiving it, or never receive it. Research on agents that can target disrupted circuits in PTSD can improve both prevention and treatment. Therefore, there is definitely room for more research on PTSD therapies and delivery methods. With the wide range of available modalities of psychotherapy and pharmacotherapy, the scheme of individualising treatment, according to the severity and personal symptom profiles of PTSD, comorbid conditions, and the use of predictive therapeutic biomarkers, can greatly enhance the efficacy of treatment. With the rapid development of new technology, research in the field of e-mental health is advancing. In light of these quick advancements, future research should concentrate on preserving a high standard of assessment of the effectiveness and acceptability of new technologies, while the evaluation of side effects and hazards should not be disregarded.

## 6. PTSD Biomarkers

A biomarker is a measurable characteristic, which can be a substance (molecular or histological), response (physiological), or structure (radiographic); it is an indicator of biological or pathological processes, or responses to exposures or interventions [170]. Recently, the identification of biomarkers for PTSD has received increasing focus [171]. PTSD biomarkers are currently used for research purposes, but they might soon assist in screening and supporting the early detection of the disorder, resulting in timely intervention and better outcomes [172]. These biomarkers could be structural changes, substances, and responses which can help assess the disease risk, diagnosis, prognosis, and response to treatment. Here, we explore the different biomarkers of PTSD.

### 6.1. Susceptibility Biomarkers

Susceptibility markers comprise those that assess the risk of developing the disorder, and are assessed before and after trauma exposure in individuals at risk [173]. Perhaps a person’s vulnerability to developing PTSD is hard to measure, and many models have been developed to explain the complexity of its evolution [174]. As discussed earlier, both pre-traumatic, peri-traumatic, and post-traumatic factors can affect disease development and progression. In terms of these, researchers have investigated several susceptibility biomarkers in the pre-traumatic and post-traumatic periods as predictors of disease. Table 2 provides a summary of the susceptibility biomarkers implicated in PTSD.

A study, the Prospective Research in Stress-Related Military Operations (PRISMO) study, investigated vulnerability markers in Dutch Armed Forces soldiers. The study included a cohort of Dutch military members deployed to Afghanistan for 4 months who experienced trauma and developed PTSD, those who did not develop PTSD, and healthy (unexposed) individuals. It was found that the number of GR in lymphocytes and monocytes before military deployment was significantly higher in soldiers who developed high amounts of PTSD symptoms after deployment which remained high for several months after deployment [175]. Moreover, T-cells’ high glucocorticoid sensitivity (GCs) (dexamethasone) before deployment was associated with high amounts of PTSD symptoms without comorbid depressive symptoms. Additionally, different patterns of GR sensitivity were associated with different presentations (e.g., severe fatigue and depressive symptoms) [176]. Furthermore, the study explored the roles of GR pathway components in disease prediction. It was found that low mRNA levels of *FKBP5* (a cochaperone modulator of receptor sensitivity to cortisol) before deployment were associated with high amounts of PTSD symptoms after deployment [177]. High glucocorticoid-induced leucine zipper mRNA levels before deployment were also associated with high amounts of PTSD symptoms post-deployment [177].

Other studies report the findings of HPA axis involvement, including one on 103 children in the USA, which concluded that polymorphisms in the CRH type 1 receptor gene were associated with PTSD development in these subjects [178]. These findings elaborate on the importance of the HPA axis, and dysregulation, and how biomarkers related to the axis can be used as disease predictors. As mentioned before, the BDNF polymorphism Val66Met also appears to increase the risk of PTSD, as it results in a decrease in BDNF expression, which obtunds conditioned fear extinction [91,182]. Studies have shown an increased frequency of the Met allele in those who develop PTSD compared to controls [93,183]. However, recent meta-analyses have found that there is no significant correlation between the Met allele and PTSD symptomatology [90,184,185].

The markers mentioned above comprise molecular ones, but other non-molecular susceptibility markers have also been investigated, especially in the post-traumatic period. Heart rate has been considered a secondary risk marker of the disease, as increased heart rates in the post-traumatic period have been associated with PTSD development [179]. Additionally, nightmares in the pre-traumatic period were associated with disease susceptibility in Dutch combat soldiers [180]. Additionally, increased skin conductance, a psychophysical marker of hyperarousal, in the immediate aftermath of trauma, was also associated with the subsequent development of PTSD [181]. As patients with PTSD tend to have higher rates of hypertension, as well as higher resting systolic and diastolic blood pressure, blood pressure is considered a candidate risk marker of PTSD [186,187]. However, a recent meta-analysis showed no association between elevated blood pressure and the subsequent development of PTSD symptoms [188].

### 6.2. Diagnostic Biomarkers

Diagnostic biomarkers are those used to assess and classify people that are already exposed to traumatic experiences, and are identified in PTSD patients in comparison to those exposed to trauma but who are disorder-free [189]. Currently, the diagnosis of PTSD is clinical, and it does not depend on its pathophysiology and underlying biological changes. Rather, it depends on the disease’s manifestation and the fulfilment of certain criteria. This is because PTSD, like all mental disorders, is complex, and phenotype manifestation differs greatly between individuals with the same level of traumatic experience [190] and even between those with similar biological and neurological activity changes. Consequently, this discrepancy might contribute to the limited applicability of such biomarkers. Therefore, biomedical biomarkers are not yet clinically used to diagnose the disorder [172], but based on scientific advances they may provide diagnostic evidence soon. In the literature, these biomarkers are classified as structural (e.g., neuroanatomical changes), peptides (e.g., monoamines and cortisol), neuroendocrine (e.g., HPA axis activity), responses (e.g., arousal, startle response), genetic and epigenetic biomarkers, and other classifiers [173,191,192]. Table 3 provides a summary of the diagnostic biomarkers implicated in PTSD.

Several studies have been conducted on the levels of monoamines in individuals with PTSD, which have indicated a rise in their levels, specifically in the case of noradrenaline, both peripherally and centrally. Hawk et al. examined the rise in catecholamines and cortisol in the urine of 55 participants who experienced serious motor vehicle accidents. They found that increased NE levels were associated with PTSD development. However, these findings were in men only, which might indicate gender differences for this biomarker [193]. Studies point to increased monoamine levels also being found in other anxiety disorders and that they are not specific to PTSD [204].

The HPA axis is the main neuroendocrine regulator in the body and is dysregulated in PTSD [205]. Several studies have concentrated on the difference in cortisol levels in individuals with PTSD, but the results were considerably variable. Meewisse et al. conducted a meta-analysis and indicated that cortisol levels are inconsistent and insignificant in patients with PTSD [206]. Interest has increased in FKBP5 and cortisol levels in response to stress. Yehuda et al. explored gene expression alterations in 35 participants who experienced the 9/11 attack on New York City. It was found that patients with PTSD had reduced levels of FKBP5 compared to controls [194]. As BDNF regulates synaptic plasticity, which is essential for fear learning and extinction, studies have suggested its role as a potential biomarker in PTSD. A systematic review and meta-analysis compared the peripheral blood levels of BDNF in patients with PTSD compared to controls without PTSD. Plasma BDNF levels were significantly higher in PTSD groups when compared to controls [202]. However, this increase in BDNF levels appears to follow a descending pattern, as BDNF levels tend to fall again in the long-term [207].

In a pilot study by Snijders et al., participants from the PRISMO study were divided into PTSD subjects, resilient subjects (trauma-exposed with no PTSD diagnosis), and non-exposed healthy controls. In this work, several miRNAs were identified as candidate diagnostic biomarkers. Five miRNAs (miR-221-3p, miR-335-5p, miR-138-5p, miR-222-3p, and miR-146-5p) were able to perfectly separate PTSD subjects from controls after adjusting for confounders [197]. Furthermore, the downregulation of miR-1246 was shown to be significant in PTSD patients compared to resilient subjects, suggesting its potential as a diagnostic biomarker [197]. Another study also suggested miRNAs as potential biomarkers, identifying a panel of nine stress-responsive miRNAs (miR-142-5p, miR-19b, miR-1928, miR-223-3p, miR-322*, miR-324, miR-421-3p, miR-463*, and miR-674*) [208].

As mentioned earlier, circuits involving the amygdala are dysregulated in PTSD. One important neuroanatomical finding in PTSD patients is amygdala over-activation [42,195]. Several studies reported increased amygdala activity in PTSD patients responding to fearful and traumatic stimuli during functional neuroimaging [209]. This hyperactivity might be due to decreased control and inhibitory signals from regulatory structures, such as the medial prefrontal cortex and the hippocampus [210]. In mentioning the hippocampus, hippocampal volume loss is a common anatomical change in patients with PTSD [196]. However, using hippocampal volume loss as a biomarker is unreliable, as it might be a direct consequence of exposure to trauma itself [211]. NPY, which is implicated in the pathophysiology of PTSD, could also serve as a diagnostic marker. Studies have shown lower NPY levels in the CSF of combat-exposed subjects with PTSD when compared to combat exposed subjects that did not develop PTSD [200,201]. Additionally, trauma exposure and PTSD are associated with diminished baseline plasma levels of NPY [198,199].

Of new interest is the role of oxytocin and its receptor expression patterns in patients with PTSD. Hofmann et al. compared serum oxytocin and oxytocin receptor mRNA (OXTR mRNA) levels at the baseline and during a Trier Social Stress Test (TSST). Serum oxytocin was found to be higher, while OXTR mRNA levels were found to be lower in the PTSD patients at the baseline compared to the healthy controls. During TSST, an increase in OXTR mRNA was markedly correlated with PTSD symptoms. It should be noted, however, that these findings apply only to the HPA axis hyporesponsive subtype of PTSD in the study [203]. However, due to the small sample size and opposing findings in other studies [212], it is still early to consider oxytocin a reliable biomarker for PTSD. Future studies should clarify its role in PTSD pathophysiology and its reliability in aiding the diagnosis.

Some psychophysical markers in patients with PTSD were also investigated. One promising candidate is skin conductance (SC), which was found to be increased in patients with PTSD [213,214]. Although resting blood pressure was found to be elevated in PTSD patients [186,187], further studies are needed to assess its feasibility as a biophysical marker. Other biomarkers suggested to help predict and diagnose PTSD include the rise in inflammatory markers and mediators (CRP, IL-2, IL-6, etc.), an increased startle response, symptoms of hyperarousal, and impaired cognitive function, among others [173,191].

### 6.3. Therapeutic Biomarkers

Therapeutic biomarkers are those that allow the prediction/monitoring of the response to the delivered treatment, and are assessed throughout the treatment process [173]. Both diagnostic and susceptibility biomarkers can contribute to the treatment process. However, a set of biomarkers that can specifically monitor treatment effectiveness and others that can predict responses to the different modalities of “stratification” have also been investigated [215,216]. Establishing a reliable and cost-effective biomarker for treatment monitoring can lead to significant improvement in PTSD management [172]. Table 4 provides a summary of the therapeutic biomarkers implicated in PTSD.

Many studies have been conducted on biomarkers predicting the response to treatment and progression. In PTSD patients, successful cognitive behavioural therapy was observed to decrease right amygdala activity while increasing right anterior cingulate cortex activity [217]. Additionally, a cerebral blood flow alteration, made evident by a difference in 99mTc-HMPAO uptake was observed between responders and non-responders to EMDR treatment. Compared to controls, patients had increased uptake in the medial temporal cortex, temporal pole, and orbitofrontal cortex. After treatment with EMDR, the uptake difference in the medial temporal cortex was not present anymore but extended to the lateral temporal cortex and the hypothalamus [218]. The medial temporal cortex is involved in memory encoding, consolidation and retrieval, and re-experiencing symptoms [222,223]. A larger rostral anterior cingulate cortex (rACC) volume was also found with a reduction in PTSD symptoms [219]. In the same study, Bryant et al. additionally found that a larger volume of rACC was present in those who responded to CBT. They also found that a greater activation of the bilateral amygdala and ventral anterior cingulate was associated with a poorer response to treatment [219]. A polymorphism in the serotonin transporter gene promoter LL 5HTTLPR was found to be associated with a better response rate to sertraline compared to the other genotypes (SS and SL) [220]. Interestingly, lower serum levels of BDNF were associated with a decrease in PTSD symptoms in chronic PTSD patients on the SSRI escitalopram [221].

As described, a significant advancement in therapeutic care will be the discovery of accurate PTSD biomarkers. Nevertheless, the ultimate therapeutic utility of biomarkers as a component of precision medicine will augment rather than replace current decision-making procedures since specific treatment needs are established through a collaborative process between patient and physician.

### 6.4. Conclusive Remarks

There are now a variety of biomarkers linked to the risks, symptoms, and course of PTSD. Despite this relationship, there is a limited prospect of employing a single marker either diagnostically or prognostically, due to the prevalent comorbidity with other mental illnesses and the limitations of studies. For instance, it is possible that decreased hippocampus volume is linked to both PTSD and comorbid depression and can act as a biomarker of the constellation of symptoms connected to both conditions. For this, biomarker panels (as opposed to the use of a single biomarker) are needed in order to maximise the specificity, sensitivity, and repeatability of diagnostic tools. Future research must examine the biological and psychological aspects of PTSD in more detail in order to meaningfully identify a combination of biomarkers that may cluster around symptoms and symptom development, for example by the use of (multi-)omics data and machine learning approaches.

## 7. Limitations and Future Directions

Since the introduction of PTSD as a diagnosis four decades ago, our understanding of the disorder has grown tremendously. However, the ability to aid recovery and enhance the quality of life of PTSD patients is still lagging behind. A lot of patients are not diagnosed timely, if at all. Although efficacious treatment regimens have been developed, many patients do not receive their treatment, and others fail to optimally respond.

Recent research on neurobiological models has given unparalleled insight into the potential underlying causes of PTSD. Yet, it is crucial to emphasise that a collection of condensed working models is unlikely to adequately describe the full intricacy of the illness. These neural models’ ability to pave the way from new pathophysiological understanding to ground-breaking treatments for PTSD may be their most important contribution. Limitations to current studies are mostly concerned with study designs and methodology, as mentioned earlier. For example, the mixed results observed in studies evaluating the HPA axis function can largely be due to traumatic exposure in the control groups being a confounder in some studies. As a result, further studies evaluating the association between trauma exposure (as opposed to PTSD) and HPA axis dysfunction are needed. The standardisation of study designs, techniques, and protocols for obtaining diurnal cortisol should be another main goal of future research. For instance, Ryan et al. suggested measuring the salivary diurnal rhythm of cortisol over a period of at least two days before and after the given intervention, as this can characterise the function of the HPA axis and the relationship between diurnal cortisol and PTSD in greater detail [224].

With the introduction of DSM-5 and ICD11, considerable modifications have been made to the diagnostic criteria and categorisation of illnesses linked to trauma and stressors. In addition to highlighting the enormous research that has gone into understanding these phenomena, the repeated revisions in diagnostic criteria and categorisation also draw attention to the difficulties that occur when evaluating disorders that are caused by traumatic experiences. Given the current understanding, there are a number of necessary next steps to comprehend how to best classify trauma- and stressor-related disorders, including, but not limited to: (1) further clarifying partial PTSD phenotypic expression and determining whether categorisation under a dimensional vs. a categorical approach would be beneficial; (2) continuing to refine assessments to ensure that traumatic experiences are thoroughly assessed and symptoms after these experiences are best captured; (3) examining possible phenotypes of stress- and trauma-related diseases in further detail. In addition, research needs to take into consideration the possible biological subtypes of HPA axis responsiveness in PTSD, as they have been found to differ significantly in symptom intensity and comorbid anxiety symptoms [225].

Because PTSD has no cure and exposure to trauma is unpredictable, it is crucial to reveal susceptibility in order to find effective resilience-building techniques and avoid PTSD from ever occurring. As only a small portion of the trauma-exposed population has PTSD, susceptibility does exist. One of the promising fields to aid this goal is the genetics and epigenetics field. New discoveries in this field can greatly aid not only the detection of susceptible individuals but also diagnosis and new routes of targeted pharmacological treatment. In a review by Al Jowf et al., we highlight the recent advancements in epigenetics and epigenomics, drawn from EWAS and GWAS studies [106]. A major limitation of these studies is the fact that a lot of these studies are preclinical studies based on animal models of PTSD. Still, a number of human cohort EWAS and GWAS studies have also been conducted, leading to candidate (epi)genetic markers [226,227,228,229]. This highlights the need for translational studies in humans that can make clinical use of these markers, which can aid in the detection of susceptibility and early diagnosis.

Biological indicators cannot yet independently validate the evaluation of PTSD, drawing a clear contrast from other medical conditions like cancer, hypertension, and autoimmune diseases that have objective biological testing procedures for diagnosis, assessing the severity of illness, and response to therapy. Instead, self-report screening tests and clinical interviews are used to diagnose PTSD rather than an identification of the underlying pathology. The growing interest in PTSD biomarkers shows significant potential and promise; however, it is currently challenging to make inferences that can be used practically from the existing fundamental and translational research on PTSD biomarkers. Once enough data are collected, machine learning approaches can help combine biomarkers in reliable, valid, and cost-effective integrated panels that can greatly enhance the prediction, diagnosis, and monitoring of the disorder.

Likewise, instead of addressing the biological etiology, treatment has mostly been restricted to symptom control and behavioural adaptation techniques. Drug development for PTSD has thus far been mostly opportunistic, based almost entirely on empirical findings using medications already licensed for other disorders. A single pharmaceutical therapy for PTSD has not yet been created as of the time of this writing. For now, future studies should pinpoint strategies for enhancing effective treatments, such as in specific populations (e.g., military personnel), for the further investigation of recommended and promising treatments, for developing strategies to individualize treatment, for maintaining patient engagement in treatment (i.e., preventing dropout), and for identifying individual factors predicting response/nonresponse. For instance, there are interventions that hold promise in improving PTSD symptoms, including repetitive transcranial magnetic stimulation (rTMS), biofeedback, exercise (e.g., yoga, and aerobic and resistance exercise) and deep brain stimulation (DBS) [230,231,232,233]. However, a lot of these studies are preclinical, while others are controversial and inadequately powered, creating a need for the further assessment for their efficacy in PTSD treatment. In the future, research on PTSD treatment should be guided by discoveries of the disruptions underpinning the development of the disorder, so that targeted and more effective interventions can be developed.

## 8. Conclusions and Perspectives

PTSD is usually associated with chronicity and disability. Although the underlying neurobiology might be elusive, several established mechanisms have been studied, which have had significant implications on management. These dysregulations include brain circuit disruption through the dysregulated release of neurotransmitters, namely NE and serotonin among many others (e.g., dopamine, GABA, and NPY), a dysfunctional HPA axis, and disordered cannabinoid and opioid activity. Although PTSD is partly attributed to these dysregulations, models such as the biopsychosocial model and the diathesis–stress model have been developed to emphasise that underlying biology is not the only contributor to the disorder, yet there is an interplay between biological (e.g., genetics, chemical changes, and organ damage), psychological (e.g., stress, mental illness, behaviour, and personality), and social factors (e.g., peers, socioeconomic status, beliefs, and culture) in the manifestations of the disease. The current treatment for PTSD involves two main modalities, namely psychotherapy and pharmacotherapy. While psychotherapy is considered the treatment of first choice, when needed, pharmacotherapy can be used as an alternative or in conjunction with psychotherapy. Preventing the disease at different time points (primary, secondary, and tertiary prevention) can significantly reduce the disease’s burden on patients’ quality of life and economic and medical burdens. Thus, the development and application of disease prevention models are of great importance. In the end, further research on susceptibility and resilience, pathophysiology, and possible targeted intervention is needed for better understanding and treatment. Over the past decade, the identification of disease biomarkers has gained more interest. Establishing reliable and cost-effective biomarkers can greatly enhance primary prevention, diagnosis, the monitoring of therapy, and the prevention of disability. None of the putative PTSD biomarkers reported so far are being used in clinical settings, which highlights the urgent need for additional studies on PTSD biomarkers with large sample sizes and for translational research strategies aiming to understand the underlying molecular causes of PTSD.

## Figures and Tables

**Figure 1 ijms-24-05238-f001:**
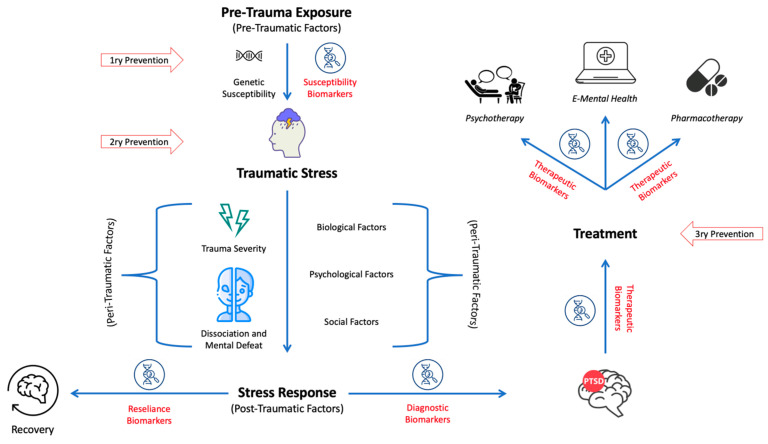
A graphical summary of the main findings of the paper. The entirety of the pre-, peri- and post-traumatic factors can be biological, psychological, or social, according to the biopsychosocial model.

**Figure 2 ijms-24-05238-f002:**
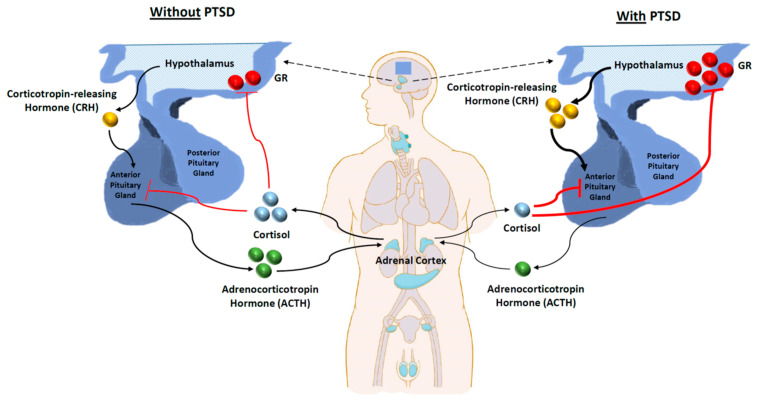
Basal activity of the HPA axis with or without PTSD. CRH secretion from the hypothalamus increases in PTSD (represented by a thicker black line). The release of ACTH from the anterior pituitary, and hence cortisol from the adrenal cortex, is decreased in PTSD (represented by a thinner black line). Cortisol’s negative feedback inhibition of the HPA axis is increased in PTSD (represented by thicker red lines).

**Figure 3 ijms-24-05238-f003:**
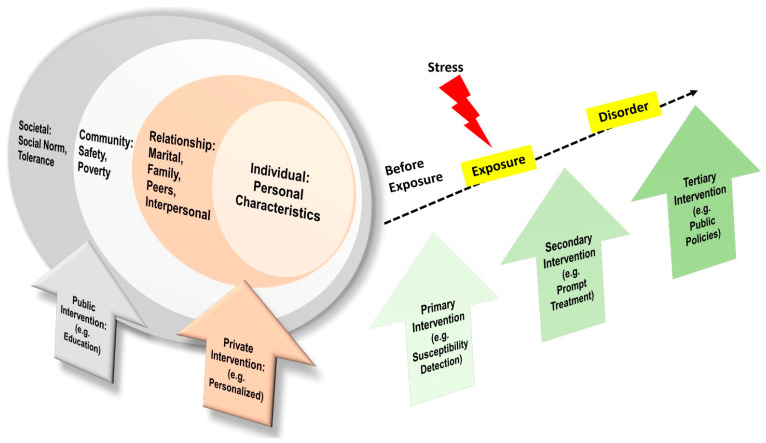
Social-ecological model for traumatic stress and related preventive interventions.

**Table 1 ijms-24-05238-t001:** Summary of the commonly used preclinical animal model in PTSD.

Model	Description	Aim
Physical:		
Foot shock stress (FSS)	A metal-rod floor is used to deliver electrical shocks, which are coupled with non-harmful cues (usually auditory) to elicit post-stress fear recall using sound in novel environments, according to the fear conditioning model	Modelling the response to inescapable stress [25]
Single prolonged stress (SPS)	Intends to result in the development of PTSD from a single traumatic experience. Rats are restrained for 2 h, subsequently forced to swim for 20 min, and then 15 min later are subjected to the ether until unconsciousness	Inducing PTSD symptoms by combining multiple, severe, and different stressors [26]
Stress-enhanced fear learning (SEFL)	Exposure to repeated foot shocks in one environment produces conditional freezing (used to assess learned fear) in response to cues associated with foot shock in a second environment	Modelling the lasting effects of traumatic stress [27]
Restraint stress (RS)	Prolonged restraint between 15 min to 2 h on a wooden board or a plastic tube	Modelling an inescapable severe psychological trauma with chronic behavioural and neurochemical alterations [28]
Underwater trauma (UWT)	Being distinct from the forced-swim test, animals are placed in deep water and are forced to swim for 30 s, before being submersed for another 30 s	Modelling an inescapable severe psychological trauma [29]
Social:		
Social defeat (SD)	Subjects are categorised as either susceptible or resilient and are exposed to and suppressed by an aggressor animal for several days. Only susceptible subjects will develop behavioural avoidance	Modelling prolonged and chronic stress as a risk of PTSD [30]
Early life stress (ELS)	From postnatal days 2–14, new-born mice are separated from their mother 1 h daily	Modelling the ability of childhood trauma to influence the development of PTSD [31]
Housing instability (HI)	Animals are frequently paired with different cohorts, after being exposed to their natural predators (e.g., cats)	Modelling the effects of housing instability in PTSD patients [32]
Psychological:		
Predator scent stress (PSS)	Animals are confronted with the scents of their natural predators (cat litter, urine, etc.)	Modelling and simulating traumatising events and trauma-related stimulus response in humans [33,34]

**Table 2 ijms-24-05238-t002:** Summary of the susceptibility biomarkers implicated in PTSD.

Susceptibility Biomarker	Findings
Number of GR in lymphocytes and monocytes	A higher number pre-trauma of GR is associated with high PTSD symptoms in soldiers after deployment [175].
Sensitivity of T cells to dexamethasone before deployment	High sensitivity pre-trauma is associated with a high amount of PTSD symptoms without comorbid depressive symptoms. Different sensitivity patterns are associated with different symptomatology [176].
mRNA levels of *FKBP5*	Low levels after deployment are associated with a high amount of PTSD symptoms [176].
Glucocorticoid-induced leucine zipper mRNA	High levels pre-trauma are associated with a high amount of PTSD symptoms post-deployment [177].
Corticotropin-releasing hormone type 1 receptor gene	Polymorphisms were associated with PTSD development [178].
Heart rate	Increased heart rates in the post-traumatic period were associated with PTSD development [179].
Occurrence of Nightmares	Higher occurrence of nightmares pre-trauma was associated with disease susceptibility in Dutch combat soldiers [180].
Increased skin conductance	Skin conductance response (SCR) within hours of trauma exposure was a predictor of chronic PTSD development [181].

**Table 3 ijms-24-05238-t003:** Summary of the diagnostic biomarkers implicated in PTSD.

Diagnostic Biomarker	Findings
Noradrenaline levels	Increased urinary noradrenaline levels were associated with PTSD development in men [193].
FKBP5 levels	Reduced FKBP5 expression in blood was found in PTSD patients [194].
Amygdala activity	Amygdala over-activation is found in PTSD patients [42,195].
Hippocampus volume	Hippocampal loss is a common anatomical change in patients with PTSD [196].
miR-138-5p overexpression miR-1246 downregulation	Plasma isolated miR-138-5p was significantly overexpressed in subjects with PTSD compared to controls, and miR-1246 was significantly downregulated in subjects with PTSD compared to resilient subjects [197].
Plasma levels of NPY	Plasma baseline levels are lower in individuals with traumatic stress exposure and PTSD [198,199].
CSF levels of NPY	Levels of NPY were lower in combat veterans with PTSD compared to veterans without PTSD and healthy controls [200,201].
Plasma BDNF level	Patients with PTSD have higher plasma levels of BDNF [202].
Oxytocin receptor mRNA levels	Blood mRNA levels of OXTR were lower in patients with hyporeactive HPA axis subtype at baseline, which increased during stress testing [203].
Others	A rise in inflammatory markers, increased startle response, symptoms of hyperarousal, and impaired cognitive function were found in PTSD patients [173,191].

**Table 4 ijms-24-05238-t004:** Summary of the therapeutic biomarkers implicated in PTSD.

Therapeutic Biomarker	Findings
Amygdala and anterior cingulate cortex activity	Successful cognitive behavioural therapy was observed to decrease right amygdala activity while increasing right anterior cingulate cortex activity [217].
Cerebral blood flow to the medial temporal cortex	A normalisation of the difference in cerebral blood flow to the medial temporal cortex after EMDR [218].
Amygdala and ventral anterior cingulate activity	The greater activation of the bilateral amygdala and ventral anterior cingulate was associated with poorer response to CBT [219].
Rostral anterior cingulate (rACC) volume	A larger rostral anterior cingulate (rACC) volume was also found in responders to CBT [219].
LL genotype of serotonin transporter gene promoter (5HTTLPR)	A polymorphism in the LL genotype 5HTTLPR was found to be associated with a better response to sertraline [220].
BDNF levels	Lower serum levels of BDNF were associated with a decrease in PTSD symptoms in chronic patients on escitalopram [221].

## Data Availability

Not applicable.

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
