# Peer review of "To Predict, Prevent, and Manage Post-Traumatic Stress Disorder (PTSD): A Review of Pathophysiology, Treatment, and Biomarkers"

_ijms, 2023, doi:10.3390/ijms24065238_

Round 1

Reviewer 1 Report

This review manuscript summarizes the current concept of findings on the etiology and disease models of post-trauma stress disorder (PTSD), pathophysiology, treatment, prevention, and biomarkers, including diagnostic and prognostic biomarkers. Available data are presented in their complexity, providing a well-balanced and insightful overview, and may be interesting to the broad range of researchers studying PTSD.  

Here are some suggestions to improve it.

Major suggestion: It lack critical thoughts from author about the Pathophysiology, Treatment, and Biomarkers of PTSD. The whole review only shows what have been done. It is better for author to add their own thoughts

Minor suggestions:

In the line 441: The author mention that E-Mental Health, there is no definition of it. Please add it.

In the line of 53-54:” including the diathesis-stress model and the biopsychosocial 53 model, for example.” Check the sentence and revise it.

In the line of 56, “alterations of the hypothalamic-pituitary-adrenal (HPA) axis dysfunction”.  Check the sentence and revise it.

In the line of 63, it better to delete the statement of “and other patients choose not to pursue treatment”, which is not related the topic of the review.

In the line of 73,” and then turns to biomarkers, including diagnostic and prognostic biomarkers” Check the sentence and revise it.

Reviewer 3 Report

Excellent paper. A lot of research and citations are given appropriately to support the information. It is a very well-written paper and is easy to follow through. The diagrams and table have also been helpful.  This is much-needed research in the world of psychiatry where bio-markers may play a vital role in both diagnosis and prognosis. I approve it as it is and do not recommend the need for any revision. Thanks so much for giving me the opportunity to review this paper. 

Reviewer 4 Report

Al Jowf and colleagues in the present review article entitled ‘To Predict, Prevent and Manage Post-Traumatic Stress Disorder (PTSD): A Review of Pathophysiology, Treatment, and Biomarkers’, investigated the current status of knowledge of neurobiological dysregulation of PTSD disorder. For this purpose, authors selected some of the most relevant evidence focusing on dysregulation of specific neurotransmitters, on dysfunction in HPA axis and on disorders in the cannabinoid and opioid activity, and finally identifying biomarkers that can greatly enhance primary prevention, diagnosis and effective treatment for this disorder.

The main strength of this paper is that it addresses an interesting and timely question, providing an overview of the current concept of findings on the etiology and disease models of PTSD, pathophysiology, treatment, prevention, including information on possible diagnostic and prognostic biomarkers of this disease. In general, I think the idea of this article is really interesting and the authors’ fascinating observations on this timely topic may be of interest to the readers of International Journal of Molecular Sciences. However, some comments, as well as some crucial evidence that should be included to support the author’s argumentation, needed to be addressed to improve the quality of the manuscript, its adequacy, and its readability prior to the publication in the present form.

Please consider the following comments:

A graphical abstract that will visually summarize the main findings of the manuscript is highly recommended.

Abstract: According to the Journal’s guidelines, the abstract should be a total of about 200 words maximum. Please correct the actual one.

I would ask the Authors to clarify the criteria they decided to use for studies’ collection in their review: they should specify the requirements used to decide whether a study met the inclusion/exclusion criteria of the review, describe whether they included a balanced coverage of all information that is actually available, whether they have included the most recent and relevant studies and enough material to show the development and limitations in this field of interest. Finally, I believe that they should briefly present results of all statistical syntheses conducted.

The objectives of this study are generally clear and to the point; however, I believe that there are some ambiguous points that require clarification or refining. In my opinion, authors should be explicit regarding how they sought to identify diagnostic biomarkers for PTSD, since this is the key aim of this review.

PTSD Biomarkers: In this section, authors focused on describing biomarkers for the presence of PTSD, which are commonly related to either dysfunction of the HPA axis, monoamine systems, heightened inflammation, genetic and epigenetic changes. In this regard, I wanted to know why the Authors decided not to include also examination of psychophysical biomarkers of PTSD, such as indicators of hyperarousal (heart rate, blood pressure, skin conductance, etc.). 

Although not mandatory, I believe that a final ‘Discussion’ section could be very useful to capture the state of art well, and in this respect, I would like to see in this section some views on a way forward, for example some good discussion on human and animal studies that have investigated the possibility of modulating aberrant fear memories, which in most of the cases characterize PTSD, by using Non-invasive brain stimulation techniques (NIBS) and optogenetics to interfere with consolidation, reconsolidation, and extinction processes. It has been demonstrated that NIBS may provide a valuable tool for interventional neurophysiology applications, modulating brain activity in a specific, distributed, cortico-cortical/subcortical network (i.e., like prefrontal-amygdala and hippocampus pathway), for example giving information on how ‘Does the human ventromedial prefrontal cortex support fear learning, fear extinction or both?’. Thus, the development of new clinical and experimental tools, such as NIBS and optogenetics allows a more accurate characterization of specific circuits (https://doi.org/10.1016/j.cub.2020.06.091) and their particular interactions within the overall fear processing network (https://doi.org/10.3389/fnbeh.2022.998714). 

I would ask the authors to include a proper and defined ‘Limitations and future directions’ section before the end of the manuscript, in which authors can describe in detail and report all the technical issues that could be brought to the surface.

Tables and Figures: According to the Journal’s guidelines, please provide a short explanatory caption for the table within the text.

References: Authors should consider revising the bibliography, as there are several incorrect citations. Indeed, according to the Journal’s guidelines, they should provide the abbreviated journal name in italics, the year of publication in bold, the volume number in italics for all the references.

I hope that, after these careful revisions, the manuscript can meet the Journal’s high standards for publication. I am available for a new round of revision of this article. 

Best regards,

Reviewer
